# OpenReview forum: "Harnessing Task Overload for Scalable Jailbreak Attacks on Large Language Models"
_ICLR.cc/2025/Conference — Submitted to ICLR 2025_

### Official Review · Reviewer_edRZ · 2024-10-26

**Soundness:** 2
**Presentation:** 3
**Contribution:** 2
**Rating:** 5
**Confidence:** 4

**Summary:**

This paper reveals that LLMs have limited computational and processing capabilities. Based on this observation, this paper proposes a new attack paradigm, which engages the LLMs in resource-intensive tasks to prevent the activation of the safety mechanism. The experimental results demonstrate the effectiveness of the method.

**Strengths:**

The paper is well written. I can clearly follow the authors’ minds.
The related works are well-studied.

**Weaknesses:**

1. The deduction of the internal mechanism for activating safety filters should be more cautious. Currently, popular LLMs can process prompts with thousands of tokens, which means that they already have a great ability to process high-load tasks. In this case, can you provide more validation about your deduction?
2. Even if there exists such a mechanism, can you claim that your task is a high-load task? How can you validate that your task activates this mechanism? This is important as the reason for the success of your attack may be you adopt an encrypt-decrypt pipeline.
3. The experiment is incomplete:
- More baselines should be included. The most related works are CodeChameleon [1] and GPT4TooSmart [2], which adopt encryption and decryption to process the input and output. CodeChameleon is quite similar to this work. You should clearly analyze it and compare your work with it.
4. Too many typos.
- line 93, load -> loading
- line 280, “both used to be” -> “are both used to be”
- line 319, “a extended” -> “an extended”
- line 500, “we” -> “We”

[1] Lv, H., Wang, X., Zhang, Y., Huang, C., Dou, S., Ye, J., ... & Huang, X. (2024). Codechameleon: Personalized encryption framework for jailbreaking large language models. arXiv preprint arXiv:2402.16717.
[2] Yuan, Y., Jiao, W., Wang, W., Huang, J. T., He, P., Shi, S., & Tu, Z. (2023). Gpt-4 is too smart to be safe: Stealthy chat with llms via cipher. arXiv preprint arXiv:2308.06463.

**Questions:**

Please refer to weaknesses

---

### Official Review · Reviewer_a8Fh · 2024-10-31

**Soundness:** 3
**Presentation:** 2
**Contribution:** 2
**Rating:** 5
**Confidence:** 3

**Summary:**

This paper proposed a scalable and transferable black-box large language model jailbreak strategy. The attacker combines a complicated *Character Map Lookup* task with the malicious prompt. By doing so, the attacker saturates the computational resources of the target large language model (LLM), thus disabling it to activate its safety mechanism and get jailbroken. This attack paradigm claims nice attack success rate (ASR) in different models and different safety benchmarks. Furthermore, the attacker can adjust the attack strength of the attack by controlling the hyperparamters (e.g., query count in the mapping task) to make it scalable to different models.

**Strengths:**

1. The proposed attack looks simple yet effective. The author's insight on jailbreak attack via occupying LLM's computational resource is fresh and insteresting to me.
2. This attack is black-box and doesn't require the attacker to optimize his/her jailbreak prompts (like GCG attacks), while it claims better ASR in most of the jailbreaking scenarios.
3. The author provides sufficient ablation studies, and shows how to control the attacker's hyperparameters (e.g., query count) to transfer the attack to different LLMs.
4. It is quite interesting to see model's safety function is surpressed during the computational resource occupying attack while the model's ability on routine instructions sometimes even improves. It could be insightful for future research.
5. In all, the paper is not hard to understand.

**Weaknesses:**

1. Some presentation details of the paper can be further polished, and there might be some potential typo errors. For example:

	In Figure 5, 8, there is no labels and value range in y-axis, which makes the reader (at least me) not able to draw enough information from the figures.

	Line 238: a injective->an injective.

	Line 254: I'm not sure if I got it wrong, but $V_{i} = concat(c_{1}, c_{n+2}, .., c_{m})$ make me confused. Why the second element be $c_{n+2}$? And shouldn't keys and values are in pairs? Why they ($K_{i}$ and $V_{i}$) ends with index n and m respectively? And if $m$ is set as 1, will be $V_{i} = concat(c_{1})$ directly?

	In Figure 15. It can be better to also put an example malicious prompt (e.g., how to make a bomb) into the template to make it more straightforward for the reader to understand. Besides, the author can show some pairs (w & w/o load task and corresponding model responses) to show the effect of the attack.

	Figure 1 (left part) is confusing to me especially when I begin to read the paper, there are all brown rectangles in it and I don't know what they mean before I read to the methodology part.

2. There are some missing experimental results in Table 2, and I think it's accessible for the author to run related experiments (e.g., GCG attack on LLama3-8b). Is there any reason for not doing them?

3. Some hypermeters (e.g., Query Count) seem not significantly/stably impact the attack success rate in involved models shown in Figure 5 (a,b looks unstable; c,d looks not significant) and Figure 7 (not significant ASR changes when query count varies, maybe you can switch another hyperparameters). This experimental result seemingly weakens the author's claim on *flexibly* controlling the attack intensity (Line 70-71). Again, no value range in Figure 5 can make me/the readers hard to judge.

**Questions:**

1. Why you come up with *Character Map Lookup* load tasks to perform the attack instead of others? Did you ever test some other load tasks?
2. The degree of preempting LLM's computational resources to attack is fresh, and if available, the authors can provide more related works (about LLM's computational resource preempting) in Sec 1 and 2 to make the readers have a better understanding of the attack's mechanism.
3. See weaknesses.

---

### Official Review · Reviewer_Pheb · 2024-11-03

**Soundness:** 2
**Presentation:** 2
**Contribution:** 2
**Rating:** 3
**Confidence:** 4

**Summary:**

This work proposes a jailbreak attack method for LLMs that exploits computational overload. By preoccupying the model with a resource-intensive task, such as a Character Map lookup, the authors effectively bypass the model’s safety protocols. The method offers flexibility in adjusting the attack’s intensity, allowing it to target models of varying sizes. The results demonstrate a high attack success rate across multiple LLMs, highlighting a potential vulnerability in the current safety designs of large models.

**Strengths:**

- This work introduces a unique strategy of saturating the model's resources as a means to bypass safety mechanisms.
- The proposed method scales across different LLMs, with controlled attack strength, making it adaptable and practical for various models.
- Extensive evaluations on multiple LLMs demonstrate the robustness and effectiveness of the method.

**Weaknesses:**

- The paper lacks a clear explanation of real-world scenarios where this specific attack method could be deployed – and if there's one – there is still a lack of discussion about basic defenses, such as detecting the usage of unusual "messy code keys".
- The computational overhead introduced by the Character Map task and finding the parameters (e.g., $\Sigma, Q, L$) are not fully addressed, leaving questions about efficiency.
- Formatting and Clarity Issues:
  - The paper’s references are inconsistently formatted (line 139), some lacking spaces before citations, and quotes usage problem (line 228).
  - Equations are not well-integrated into the text, and some lack clarity in their necessity or impact on the analysis. Function 3 lacks strictness, and whether "S policy" is a set of token strings. Simpler language could make these definitions clearer. Function 5 introduced $\mathbb{c}$ – which is not necessary – it's not related to ASR, not to say $\Sigma, Q, L$.
  - Figure 5 lacks a y-axis, and Figure 6’s ASR metrics from 0-100 appear inconsistent compared to other tables and figures.

**Questions:**

- How would the authors envision deploying this attack method practically? Are parameters like $\Sigma$, $Q$ and $L$ adjustable on-the-fly, or do they require preset configurations?
- For keys in the Character Map, is there a length constraint (single token vs. multiple tokens), or does length not impact performance?
- What is the exact attack strength level used in Table 2's main experiments?
- I don't quite get the conclusion in Sec. 5.2, i.e. how to calibrate the attack strength? The hidden relationship between model performance, ASR, $\Sigma, Q, L$ is complex.

---

### Official Review · Reviewer_W8tx · 2024-11-03

**Soundness:** 2
**Presentation:** 3
**Contribution:** 2
**Rating:** 5
**Confidence:** 4

**Summary:**

In this paper, the authors propose a method for conducting jailbreak attacks on LLM by strategically overloading their computational resources with preliminary load tasks. The preliminary load tasks are constructed character mapping lookup  task.

**Strengths:**

- The idea of using task overload to increase the success rate of jailbreaking is interesting.
- They did a good formalization for presenting the attack.
- The ablation study is sufficient.

**Weaknesses:**

- The main hypothesis of the attack is: saturating the model's processing capability can prevent the activation of safety protocols. Why? The safety alignment is mainly done on the parameters of the LLMs, especially all the experiments are done on open-sourced LLMs. There seems no evidence for the above hypothesis to be convincing.
- Both PAIR and GCG are evaluated on commercial models, which is lacked in this paper. Therefore, to claim the task overloading approach is a more threatening attack is unfair.
- What is the substantial difference of this attack from jailbreaking attacks based on, e.g., word puzzles [1]. I think to propose a novel jailbreaking attack, there should be more insights (which is grounded in the underlying mechanism of LLMs) on why the attack is different from a trick.

[1] https://arxiv.org/html/2405.14023

**Questions:**

See the weakness part above.

**Details Of Ethics Concerns:**

N/A.

---

### Author Response · Authors · 2024-11-26
**Thanks for reviewers and editors**

We sincerely appreciate the valuable feedback provided by all reviewers and the editor on our paper. We are grateful for the time they invested in the review process. We think that there are still areas in the paper that can be improved, we have decided to revise and enhance it further to ensure a higher quality.

---

### Meta-Review · Area_Chair_2xJW · 2024-12-20

**Metareview:**

The paper introduces a jailbreak attack that exploits computational resource overload through character map lookup tasks. Based on the paper’s presentation, the method outperforms several baseline methods, such as GCG and FAIR.

All reviewers find the paper well-structured and easy to follow, commending the innovative concept of using computational resource saturation as a means to bypass safety mechanisms. Reviewer (Pheb, a8Fh) highlights the scalability of the proposed method.

However, the reviewers have also raised several concerns. Reviewer W8tx questions the validity of the main hypothesis underlying the method. Additionally, multiple reviewers (Pheb, a8Fh, edRZ) mention that the paper lacks clarity in its presentation. Several reviewers (W8tx, edRZ, and a8Fh) also point out the absence of adequate baseline comparisons. Reviewer Pheb questions the practicality and efficiency of the proposed attack in real-world scenarios. Reviewer a8Fh highlights issues with the choice of hyperparameters. Reviewer edRZ questions the contribution of high-load tasks to the overall success of the attack.

The reviewers ' concerns remain unaddressed since the authors did not participate in the rebuttal pase. After careful consideration of all factors, the AC recommends rejecting the paper.

**Additional Comments On Reviewer Discussion:**

The authors did participate in the rebuttal, and the reviewers' concerns remain unaddressed.

---

### Decision · Program_Chairs · 2025-01-22

Reject